# Use of Collagen Membrane in the Treatment of Periodontal Defects Distal to Mandibular Second Molars Following Surgical Removal of Impacted Mandibular Third Molars: A Comparative Clinical Study

**DOI:** 10.3390/biology10121348

**Published:** 2021-12-18

**Authors:** C. Burnice Nalina Kumari, Thiagarajan Ramakrishnan, Pradeep Devadoss, Rajaram Vijayalakshmi, Khalid J. Alzahrani, Mazen A. Almasri, Manea Musa. Al-Ahmari, Hajar Saeed Al Dira, Malath Suhluli, Ashok Kumar Bhati, Zeeshan Heera Ahmad, A. Thirumal Raj, Shilpa Bhandi, Shankargouda Patil

**Affiliations:** 1Department of Periodontology and Implantology, Faculty of Dentistry, Meenakshi Academy of Higher Education and Research, Chennai 600095, India; ramkidentist1961@gmail.com (T.R.); drvijaya.perio@madch.edu.in (R.V.); 2Department of Oral and Maxillofacial Surgery, Faculty of Dentistry, Meenakshi Academy of Higher Education and Research, Chennai 600095, India; drpradeep.omfs@madch.edu.in; 3Department of Clinical Laboratories Sciences, College of Applied Medical Sciences, Taif University, Taif 11099, Saudi Arabia; ak.jamaan@tu.edu.sa; 4Department of Oral Maxillofacial Surgery, King Abdulaziz University, Jeddah 21589, Saudi Arabia; malmasri@kau.edu.sa; 5Department of Periodontics and Community Dental Sciences, College of Dentistry, King Khalid University, Abha 61421, Saudi Arabia; abudanahmm@gmail.com; 6College of Dentistry, Jazan University, Jazan 45142, Saudi Arabia; hajar24688@gmail.com (H.S.A.D.); ml3th2538@gmail.com (M.S.); 7Division of Periodontics, Department of Preventive Dental Sciences, College of Dentistry, Jazan University, Jazan 45142, Saudi Arabia; gums_ashh@yahoo.com; 8Dental College Hospital, King Saud University Medical City, Riyadh 11362, Saudi Arabia; aheera@ksu.edu.sa; 9Department of Oral Pathology and Microbiology, Sri Venkateswara Dental College and Hospital, Chennai 600130, India; thirumalraj666@gmail.com; 10Department of Restorative Dental Science, College of Dentistry, Jazan University, Jazan 45142, Saudi Arabia; shilpa.bhandi@gmail.com; 11Division of Oral Pathology, Department of Maxillofacial Surgery and Diagnostic Sciences, College of Dentistry, Jazan University, Jazan 45142, Saudi Arabia

**Keywords:** collagen membrane, extraction, impacted, mandibular third molar, periodontal defect

## Abstract

**Simple Summary:**

Regeneration of the wounded/lost tissues post-surgery often requires a meticulous treatment plan. The present study assessed the use of a collagen membrane to aid in the regeneration of lost periodontal tissues following extraction of the tooth. The results show that the collagen membrane promoted wound stabilization, as indicated by improvement in all assessed clinical parameters. Thus, collagen membrane can be used regularly in clinical practice post-extraction to augment wound healing.

**Abstract:**

The study aims to assess the efficacy of using collagen membrane in the treatment of distal periodontal defects of mandibular second molars following the removal of mesioangularly or horizontally impacted mandibular third molars surgically. Forty sites in twenty patients with bilaterally impacted mandibular third molars (mesioangular or horizontal) were considered for the study. In 20 test sites (Group A), after surgical removal of the mandibular third molar, a resorbable collagen membrane barrier was placed on the distal aspect of the mandibular second molar to cover the post-surgical bone defect. In the other control 20 sites (Group B), the same surgical procedure was repeated without placing any membrane barrier. The clinical parameters recorded were Oral Hygiene Index Simplified (OHI-S), Probing pocket depth (PPD), Clinical attachment level (CAL), and radiographic assessment of alveolar bone level (ABL). OHI-S score of most of the patients was observed to be satisfactory. Group A was observed to achieve a statistically significant reduction in PPD, CAL, and ABL gain compared to Group B. The improvements indicated that the use of collagen membrane facilitates early wound stabilization and promotes primary closure of the defect. This recovery is achieved through its unique property to assist fibrinogenesis over osteoconduction. Further longitudinal studies are needed to confirm the present findings.

## 1. Introduction

Mandibular third molars vent out in the terminal end of the lower dental arch and are the most recurrently wedged [1,2]. Conceivable contributing elements to the impaction incorporate deficient dental arch space and erratic eruption paths. Impacted third molars present with various clinical conditions such as pericoronitis, caries, periodontitis, root resorption of the second molar, cystic changes, edema, alveolitis, orthodontic problems, or TMJ dysfunction [2,3,4,5]. Notwithstanding, injuries related to an affected mandibular third molar may now and then be asymptomatic. An example is a confined periodontal issue on the adjoining second molar that is related to extraction of affected or incompletely emitted mandibular third molar [6,7,8]. Literature on periodontal wound healing has brought about the improvement of the treatment methodology using guided tissue regeneration (GTR) [9,10]. In this treatment, a barrier membrane is placed to cover the periodontal defect so that the gingival tissues (gingival epithelium and connective tissue) are kept away from reaching the root surface during recuperating. Simultaneously, space is shared between the obstruction and the root, permitting periodontal tendon (PDL) cells to repopulate the exposed bit of the root and produce another connective tissue connection. A resorbable collagen film has been presented for use in GTR treatment. The strength of the developing fibrin clot is a prerequisite for legitimate healing [11]. The use of a membrane is expected to promote wound stabilization and defend the adhering clot from tensile forces onto the external portion of the membrane [11]. This barrier allows the body to regenerate periodontal ligament, cementum, and bone [12]. Thus, RCMs could be successfully utilized in clinics to treat localized periodontitis and increasing patient comfort. The primary aim of the study was to assess the clinical efficacy of collagen membrane in treating periodontal defects on the distal aspect of mandibular second molars following trans alveolar extraction of impacted mandibular third molars. The secondary aim is to compare the periodontal defect resolution on the distal aspect of mandibular second molar with and without the placement of collagen membrane after transalveolar extraction of impacted mandibular third molars.

## 2. Materials and Methods

A total of 20 patients who reported major complaints of pain in the lower back tooth region because of bilaterally impacted mandibular third molars were selected for the study. The study was conducted at the Department of Periodontology at Meenakshi Ammal Dental College and Hospital, Chennai. 

The study samples consisted of 40 sites in 20 patients (10 male and 10 female) in the age group of 25 to 40 years. The pre-operative assessment for the impacted tooth included clinical and radiological assessment using intraoral periapical radiographs to assess the depth and type of tooth impaction. Radiovisiography (RVG) was also considered to standardize the alveolar bone measurements.

Study Design: Patients with bilaterally impacted mandibular third molar teeth were selected for the study. Before extraction of the impacted molars, the sites were randomly divided into Group A (Test) and Group B (Control) in a split-mouth design.

Group A (Test Site): In 20 sites, after surgical removal of the mandibular third molar, a resorbable collagen membrane barrier (Healiguide^TM^, Chennai, TamilNadu) was placed on the distal aspect of the mandibular second molar to cover the post-surgical bone defect. The flap was then replaced and sutured to cover the barrier. 

Group B (Control Site): The same surgical procedure was repeated for the other 20 sites without the membrane barrier.

Inclusion Criteria: Patients 25–40 years of age were considered for the study who exhibited good overall health evident from the pre-operative history. Additionally, patients who also presented with bilateral impacted mandibular third molars with mesioangular or horizontal inclination and the alveolar crest located 4 mm apical to Cementoenamel Junction (CEJ) on the distal aspect of the mandibular second molar were also included.

Exclusion Criteria: Patients with a background marked by atopy that as per the examiner would preclude periodontal surgical procedure and the utilization of antibiotics.
Patients with poor oral hygiene (OHI-S score > 3 are considered poor).Patients who have systemic conditions that as per the examiner would preclude surgery.Patients diagnosed with acute infectious lesions in areas where the surgery must be done.Smokers.Patients with a habit of alcohol or drug abuse.

Clinical Parameters: The clinical parameters considered for the study were Oral Hygiene Index-Simplified (OHI-S), Probing Pocket Depth (PPD), and Clinical Attachment Level (CAL). OHI-S score was calculated by visual examination of supra and subgingival calculus by using an explorer on index teeth and were assigned scores from 0–3 separately for debris and calculus, based on the following criteria:0 = No debris/stain present;1 = soft debris covering not more than 1/3rd of the tooth surface being examined or the presence of extrinsic stains without debris regardless of the surface area covered;2 = soft debris covering more than 1/3rd but not more than 2/3rd of the exposed tooth surface;3 = soft debris covering more than 2/3rd of the exposed tooth surface.

Probing pocket depth is measured by inserting the probe parallel to the vertical axis of the tooth and walked circumferentially around each surface of the tooth to detect the area of deepest penetration. Clinical attachment loss is determined by subtracting the level of attachment from the depth of the pocket. If both are same, the loss of attachment is zero.

Radiographic Parameter: Alveolar bone level (ABL) was measured radiographically at the distal surface of mandibular second molar using Radiovisiography (RVG) and was imported to Sopro^®^ software 2.40, which yields an accuracy of 0.1 mm.

Presurgical Protocol: An informed consent was taken from the patients before the start of the treatment.

The preoperative clinical and radiographic parameters were measured (Figure 1a,g and Figure 2a,e). Each patient was prepared for the surgery with the first stage of periodontal therapy, which included scaling and root planning, along with instructions to maintain oral hygiene and occlusal adjustment wherever required. 

After the completion of phase I therapy, the sites for surgery were assigned to the study groups (A or B) in a randomized fashion and scheduled for surgery.

Surgical Technique

Preparation of the Recipient Bed: The surgical area was prepared and anesthetized using 2% Lignocaine HCl containing 1:200,000 epinephrine. A trans alveolar extraction procedure was performed in both groups by the same oral surgeon (Figure 1c and Figure 2c).

Procedure: Trans alveolar extraction was performed by assessing the Winters WAR lines through an intraoral periapical radiograph. Using a no.15 surgical BP blade, a Wards incision starting from the crevices of the second molar extending distally along the external oblique ridge was made. A mucoperiosteal flap was raised, and the impacted tooth was exposed (Figure 1b and Figure 2b). Using a 702 surgical straight fissure bur, buccal and distal guttering was prepared. Sectioning of the tooth was performed depending on the tooth position and interference from the second molar. The tooth was luxated using a straight or Coupland elevator and delivered with lower molar extraction forceps.

Following the extraction, the exposed root surfaces on the distal aspect of the second molar were gently curetted with sharp curettes (Figure 1c and Figure 2c). In group A sites, collagen membrane (Healiguide^TM^) was placed on the distal aspect of the mandibular second molar. It was trimmed according to the size required at the recipient site and was adapted to the root surface with the help of Ethicon 4-0 resorbable suture material (Vicryl) (Figure 1d,e). The flap was then repositioned to the membrane entirely and secured with 4-0 black silk sutures (Figure 1f). In group B sites, the flap was repositioned and secured with 4-0 black silk sutures (Figure 2d) without the collagen membrane.

All the patients were prescribed Capsule Amoxycillin 500 mg, Tab Metrogyl 400 mg, and Tab Imol (combination of ibuprofen 400 mg and paracetamol 500 mg) thrice daily for 5 days. A mouth rinse of 0.2% Chlorhexidine was also prescribed for two weeks after the surgery. No clinical or radiographic parameters were measured on the day of surgery to avoid further discomfort to the patient. 

Post-operative Instructions: Post-operative instructions were first explained to the patient, and handouts of these instructions were given as reinforcement. Patient instructions included the following: avoid vigorous exercise, avoid very hot food or drink for the rest of the day, and use a soft-bristle toothbrush for effective plaque control at the surgical area by using a coronally directed roll technique. 

Follow-Up Visits: On day 7 post-operative, the sutures were removed and the treated one was assessed for any potential complications. 

OHI-S and ABL measurements using RVG were assessed only on the day of suture removal. These scores were taken as baseline scores (day 0) (Figure 1h and Figure 2f). 

The surgical sites were evaluated on follow-up visits post-operatively at day 90 and day 180. All the clinical parameters and radiographic parameters were again recorded. (Figure 1i,j and Figure 2g,h)

For clinical and statistical analysis, pre-operative measurements before phase I therapy and the measurements at baseline (day 0), 3rd, and 6th months were considered.

Statistical Analysis: Mean and standard deviation for both groups at pre-operative day, baseline (day 0), day 90, and day 180 were estimated. Wilcoxon Signed-Rank Test was used for the statistical test after adjusting the *p*-values for multiple comparisons by using the Bonferroni correction method. The data were confirmed using statistical package for social science (SPSS) software. Significance was set at *p* < 0.05.

## 3. Results

Three out of twenty patients had failed to resume the treatment or evaluation due to post-operative bleeding (1 patient), paresthesia of tongue (1 patient), or post-operative pain, and with swelling and fever (1 patient), and hence were excluded from the study. The mean reduction of OHI-S score from pre-operative day to day 0, day 90, and day 180 was found to be statistically significant (*p* < 0.0001) (Table 1).

The mean PPD (Table 2) at pre-operative days between Group A and B was not statistically significant (*p* = 0.14). However, the mean PPD reduction was observed to be significantly increased in Group A than B on day 90 (*p* = 0.01) and day 180 (*p* = 0.001).

The mean CAL (Table 3) on a pre-operative day in Group A was statistically significant (*p* = 0.02) compared to Group B. On day 90 and day 180, the mean CAL gain was higher in Group A than in Group B. The mean change of ABL (Table 4) was observed to be significantly more in Group A than B (*p* = 0.003).

## 4. Discussion

The extraction of the affected third molar may lead to bone loss, increase in probing pocket depth, and introduction of the cementum on the distal aspect of the root surface of the nearby second molar. All these factors influence the long-term prognosis of the subsequent molar. Bony defects in older people may continue prompting periodontal deformity with loss of periodontal structures distal to affected mandibular third molars [13,14]. Careful mediations might be consequently needed to improvise or to dispose of such periodontal defects. There is a danger that a periodontal bony defect at the distal aspect of the second mandibular molar will not heal after careful expulsion of not many sorts of affected impacted third molars in patients over 24 years of age [7,8,14]. A regenerative approach is ideal as it can remake the lost periodontium by the formation of new attachment and opposite the desolates of the disease process [15]. Periodontal regeneration refers to the process of regeneration of cells and fibers and remodelling of lost periodontal structures [16,17,18]. Bioabsorbable collagen membrane [19] has shown notoriety because collagen invigorates platelet attachment, improves the fibrin linkage, and is a chemotactic action for fibroblasts. It additionally represses apical movement of the epithelium and balances out the wound [20]. Collagen additionally empowers the epithelial cells and autogenous connective tissue to join and move over its surface [21]. Bioabsorbable films, especially hyaluronic- and collagen-corrosive films, may promoted bone regeneration through their action on osteoblasts [22]. Stavropoulos et al. [23] and Eickholz et al. [24] comprehensively elucidated the long-term effect of GTR membranes on intrabony defect. Collagen is furthermore resorbed in the tissues utilizing the catabolic processes and thus is eventually substituted by new collagen, signifying that the collagen membrane can act as a barrier for GTR and contribute to the volume of collagenous tissue at the surgical site [19,20,25]. Healiguide^TM^ is a bio-resorbable collagen membrane derived from the bovine deep flexor (Achilles) tendon. It is made up of type I collagen and used as a GTR membrane in various periodontal procedures including the treatment of intrabony defects. This membrane is semi-occlusive, incorporated into surrounding tissue, and is resorbed completely in 12 weeks. It also has additional charge modification and slight calcification in the collagen membrane, which helps in achieving better guided tissue regeneration than other collagen membranes. This property makes it unique and different from other native collagen membrane and is known to facilitate fibrogenesis over osteoconduction. The porosity is less than the penetrable size of epithelial cell during the initial healing period [26]. Jim-Charm Kim [27] expressed that the position of collagen membrane after extraction of impacted third molar diminished the complications post-operatively at the early stage and improved the initial healing of gingival soft tissues and periodontal defects. Isidoro Cortell-Ballester [28] hypothesized that the utilization of resorbable collagen membrane after careful extraction of mesioangular third molars or evenly affected mandibular third molars stimulated bone healing, improving the clinical attachment level and bone-defect fill distal to the mandibular second molar. In like manner, it additionally decreased the distal probing pocket depth and brought about quicker recuperation. Thus, resorbable collagen membrane placement after careful extraction of an affected mandibular third molar was suggested because it forestalls periodontal bony defects after a mandibular third molar surgical procedure. The management of soft tissue and osseous defects distal to second molars because of the careful extraction of affected third molars could be a challenge. The antimicrobial treatment gives clinical advantages during the treatment of bony defects by forestalling bacterial contamination during the recuperating period as per a meta-examination done by Herrera D et al. [29]. A twice-a-day mouth rinse (0.2% chlorhexidine gluconate mouth rinse) was advised for 14 days [30]. 

The effect of the periodontal defects on the distal aspect of the mandibular second molar after the extraction of an impacted third molar with and without the placement of the GTR membrane has been studied previously [31]. The results showed significant improvements in PPD reduction, CAL gain, and ABL gain in both the test and control sites with no significant differences [32,33,34]. Stability of the maturing fibrin clot is required for appropriate wound healing. The use of a membrane is speculated to augment wound stabilization and defend the adhering clot from tensile forces onto the external portion of the membrane [11]. In our study, out of the 17 patients, 15 showed an improvement in the alveolar bone formation as reported previously [31,35,36] in the test site (4.4 ± 0.6 mm) as compared to the control site (6.1 ± 1.1 mm). The remaining two patients presented a significant amount of alveolar bone level formation in both the test and control sites, although statistical significance was not observed [31,32,33,34,36,37]. The reduction in the OHI-S score from the pre-operative level to the end of the study period was achieved by the adequate plaque control program performed by the patients. Inadequate plaque control after the extraction of impacted third molars results in the predisposition of a persistent localized periodontal problems [7,8]. A lower brushing frequency before surgery and during the first post-operative week after surgery was reported to be related to greater pain scores [38]. Plaque accumulation in the defective area was one of the significantly associated consequences and was explained to the patients in the study. At each visit, oral hygiene techniques were advised through motivational demonstrations. 

Enhancement of autogenous connective tissue, which leads to the attachment and migration of epithelial cells on the surface of the tissue, could be a contributing factor for the increased PPD in Group A [21]. Platelet attachment enhanced fibrin linkage, and chemotaxis for fibroblasts is also established by collagen [39]. The potential for periodontal regeneration is increased with the use of collagen membrane as it is exceedingly compatible and integrates well with the connective tissue wall [22]. The extent of attachment loss was found to be lower in Group A compared to Group B. This decreased attachment loss in Group A could also be attributed to collagen, which encourages autogenous connective tissue [21]. Moreover, Alpar et al. [40] indicated that the collagen barrier is exceptionally compatible and might easily integrate into connective tissue walls, thereby increasing the potential for periodontal regeneration as compared to other membranes. In the present study, for the assessment of alveolar bone levels (ABLs), standardized radiographic measurements for the selected patients were recorded using direct digital radiography, and the images were imported to Sopro^®^ software 2.40(Satelec), which yields an accuracy of 0.1 mm, to measure the distance between CEJ to the base of the defect. The direct digital radiography provided added advantages of being more sensitive, allowed or the acquisition of immediate images and adjustment of grayscale images, and reduced the exposure dose by approximately 91–96% [41]. 

The mean ABL gain was found to be greater in Group A, which was statistically significant (*p* < 0.0001). This is in concordance with the previously reported studies by Leung et al. [38] and Aimetti et al. [35,37]. The bioabsorbable collagen membrane is known to considerably amplify collagen synthesis and alkaline phosphatase activity. Collagen also increases the secretion of TGF-beta_1_, a growth factor involved in bone remodeling. The bioabsorbable collagen membrane, therefore, promotes bone regeneration, which increases alveolar bone formation [22]. Bioabsorbable collagen membranes in guided tissue regeneration treatment of intrabony defects distal to the mandibular second molar have been reported to obtain similar probing depth reductions and clinical attachment level gain than non-resorbable ePTFE membranes after third molar extraction [42].

Grafting of osseous defects with xenograft has also been established, which caused a significant reduction in the probing depth, clinical attachment level, and bone fill [43]. Exposure of membranes is a well-known impediment associated with GTR-based procedures [11]. However, in this study, GTR-treated sites were not presented by membrane exposure. This was possible due to steps taken to alleviate potential tension, allowing tension-free primary coverage. A shortcoming of this study is the lack of blinded evaluations. A conscious attempt has been made to minimize this effect. However, all surgical procedures and evaluations were performed by only one surgeon. Additionally, the results were analyzed for a shorter period (till day 180). Nevertheless, prolonged follow-up is essential to provide more conclusive remarks on the study. Although advanced biomaterials and membranes have evolved recently, collagen membrane remains beneficial with regard to healing, increased biocompatibility, patient morbidity, and post-operative complications.

## 5. Conclusions

The results of the study showed that, in comparing Group A and Group B, group A showed better probing pocket depth reduction and clinical attachment level gain when compared to Group B. However, the radiographic bone fill that remained unchanged between groups during the 90th day showed a significant decrease in group A at the 180th da. This could be because the regeneration takes 6 months’ time duration. The utilization of collagen membrane (Healiguide^TM^) in the treatment of periodontal distal defects of mandibular third molars led to more favorable clinical and radiographic outcomes as compared to the control group without the membrane. Further longitudinal studies are obligatory to confirm the present findings.

## Figures and Tables

**Figure 1 biology-10-01348-f001:**
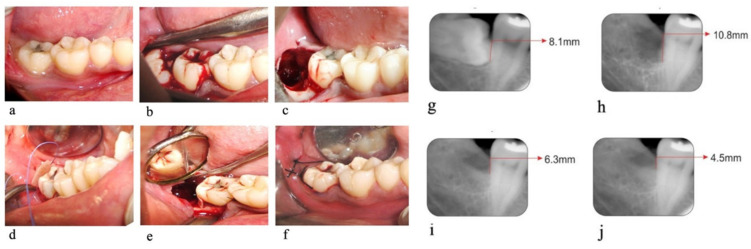
Test Site. (**a**) Pre-operative photograph, (**b**) exposed impacted tooth, (**c**) socket immediately after extraction, (**d**,**e**) collagen membrane placed on the distal aspect of the second molar, (**f**) primary closure of the socket. (**g**) Pre-operative Radiograph, (**h**) radiographs on the day of suture removal, (**i**) radiographs on 90th day, (**j**) radiographs on 180th day.

**Figure 2 biology-10-01348-f002:**
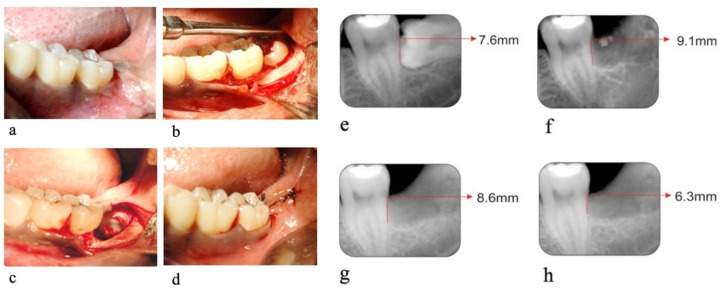
Control Site. (**a**) Pre-operative photograph, (**b**) exposed impacted tooth, (**c**) Extraction socket on the distal aspect of the second molar, (**d**) primary closure of socket. (**e**) Pre-operative radiograph, (**f**) radiographs on the day of suture removal, (**g**) radiographs on 90th day, (**h**) radiographs on 180th day.

**Table 1 biology-10-01348-t001:** Comparison of mean oral hygiene index-simplified (OHI-S) score between different time points.

Time Points	Mean ± S.D.	Difference	*p*-Value
		Mean ± S.D.	
Pre–op	2.5 ± 0.7	1.9 ± 0.6	<0.0001 (Sig.)
Day 0	0.6 ± 0.2		
Pre–op	2.5 ± 0.7	1.7 ± 0.6	<0.0001 (Sig.)
Day 90	0.8 ± 0.2		
Pre–op	2.5 ± 0.7	1.3 ± 0.5	<0.0001 (Sig.)
Day 180	1.2 ± 0.4		
Day 0	0.6 ± 0.2	−0.2 ± 0.1	0.006 (Sig.)
Day 90	0.8 ± 0.2		
Day 0	0.6 ± 0.2	−0.6 ± 0.3	<0.0001 (Sig.)
Day 180	1.2 ± 0.4		
Day 90	0.8 ± 0.2	−0.4 ± 0.3	0.006 (Sig.)
Day 180	1.2 ± 0.4		

**Table 2 biology-10-01348-t002:** Comparison of mean values of probing pocket depth (PPD in mm) at different time points between two study groups.

Time Points	Group A (Test)	Group B (Control)	Intergroup *p*-Value
	Mean ± S.D.	Mean ± S.D.	
Pre-OP	6.9 ± 0.9	6.5 ± 0.9	0.14 (N.S.)
Day 90	6.9 ± 0.8	7.8 ± 1.0	0.01 (Sig.)
180th day	4.0 ± 0.7	5.7 ± 1.1	0.001 (Sig.)
Mean Change from Pre-op to Day 90	−0.1 ± 0.7	−1.3 ± 0.8	0.001 (Sig.)
Intragroup ***p*-value**	1.00 (N.S.)	0.003 (Sig.)	
Mean Change from Pre-op to Day 180	2.9 ± 1.0	0.8 ± 1.2	0.001 (Sig.)
Intragroup ***p*-value**	<0.0001 (Sig.)	0.07 (N.S.)	
Mean Change from Day 90 to Day 180	2.9 ± 0.7	2.1 ± 0.7	0.004 (Sig.)
Intragroup ***p*-value**	<0.0001 (Sig)	<0.0001 (Sig.)	

**Table 3 biology-10-01348-t003:** Comparison of mean values of clinical attachment level (CAL in mm) at different time points between two study groups.

Time Point	Group A (Test)	Group B (Control)	Intergroup *p*-Value
	Mean ± S.D.	Mean ± S.D.	
Pre-op	7.4 ± 0.9	6.9 ± 0.8	0.02 (Sig.)
Day 90	8.4 ± 1.0	8.9 ± 1.1	0.01 (Sig.)
Day 180	4.2 ± 0.7	5.9 ± 1.1	0.001 (Sig.)
Mean Change from Pre-op to Day 90	−0.9 ± 1.0	−1.9 ± 1.3	0.002 (Sig.)
Intragroup ***p*-value**	0.01 (Sig.)	0.003 (Sig.)	
Mean Change from Pre-op to Day 180	3.2 ± 1.0	1.1 ± 1.3	<0.0001 (Sig.)
Intragroup ***p*-value**	<0.0001 (Sig.)	0.01 (N.S.)	
Mean Change from Day 90 to Day 180	4.1 ± 0.9	3.0 ± 0.7	0.005 (Sig.)
Intragroup ***p*-value**	<0.0001 (Sig)	<0.0001 (Sig.)	

**Table 4 biology-10-01348-t004:** Comparison of mean values of alveolar bone level (ABL in mm) at different time points between two study groups.

Time Point	Group I	Group II	Intergroup*p*-Value
Mean ± S.D.	Mean ± S.D.
**Pre-op**	7.6 ± 0.9	7.2 ± 1.0	0.04 (Sig.)
**Day 0**	9.9 ± 1.1	10.0 ± 1.4	0.79 (N.S.)
**Day 90**	6.8 ± 0.6	7.1 ± 0.9	0.13 (N.S.)
**Day 180**	4.4 ± 0.6	6.1 ± 1.1	0.001 (Sig.)
Change from Pre-op to Day 0	−2.2 ± 0.4	−2.8 ± 1.1	0.06 (N.S.)
IntraGroup **(*p*-value)**	<0.0001 (Sig.)	<0.0001 (Sig.)	
Change from Pre-op to Day 90	0.8 ± 0.8	0.1 ± 0.9	0.005 (Sig.)
IntraGroup **(*p*-value)**	0.02 (Sig.)	1.00 (N.S.)	
Change from Pre-op to Day 180	3.3 ± 1.1	1.2 ± 0.7	<0.0001 (Sig.)
IntraGroup **(*p*-value)**	<0.0001 (Sig.)	<0.0001 (Sig.)	
Change from Day 0 to Day 90	3.1 ± 0.8	2.9 ± 1.5	0.52 (N.S.)
IntraGroup **(*p*-value)**	<0.0001 (Sig.)	<0.0001 (Sig.)	
Change from Day 0 to Day 180	5.5 ± 1.2	3.9 ± 1.3	0.003 (Sig.)
IntraGroup **(*p*-value)**	<0.0001 (Sig.)	<0.0001 (Sig.)	
Change from 90th to Day 180	2.5 ± 0.7	1.1 ± 0.9	0.001 (Sig.)
IntraGroup **(*p*-value)**	<0.0001 (Sig.)	0.006 (Sig.)

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
