# Peer review of "Use of Collagen Membrane in the Treatment of Periodontal Defects Distal to Mandibular Second Molars Following Surgical Removal of Impacted Mandibular Third Molars: A Comparative Clinical Study"

_biology, 2021, doi:10.3390/biology10121348_

Round 1

Reviewer 1 Report

In the introduction and discussion, there are several examples of ponctuation lack before references, and space between words and references is not well defined.  Also, the first sentence of the introduction should be an unquestionable affirmation; therefore there is a need of complementing this first sentence with more references.

In papers standard writing using SI units, the unit is separated of the number by a space. This rule is only an exception for angles degrees and time nomenclature using prima.

The first time that cementoenamel junction appears, there is no reference to its abbreviation (CEJ). Then in line 103, the abbreviation was written without mention before the term. 

Replece inside marign Exclusion criteria.

Line 151: Postoperative Instructions or Post‐operative instructions?

How to integrate the sentence of lines 152/153 with the information of lines 154, 155 and 156?

Please try to standardize pre-operative or preoperative.

Shouldn't lines 184 to 186, be placed on 172 to 174?

I feel the need to been described how is evaluated the mean oral hygiene index‐simplified (OHI‐S) score. is this score in a range of 0 to 5, 0-10, 0-20 ? Which are the criteria of oral hygiene index‐simplified (OHI‐S) score?

I understand that discussion brings some topics that should be carried out in introduction, as example do a revision on similar membranes existing in market and why you choose Healiguide?

Also in conclusions, I consider it should point out parcial conclusions of the results, in order to offer to the readers a board perspective of the main conclusions of the work.

I understood that alveolar bone level (ABL in mm) is measured on the radiographic images, but how is measured clinical attachment level (CAL in mm)? And probing pocket depth (PPD in mm). Please explain OHI-S, CAL and PPD, procedure. The only procedure I understood was ABL, which was correctly explained.

Author Response

Reviewer 1:

  1. In the introduction and discussion, there are several examples of ponctuation lack before references, and space between words and references is not well defined.  Also, the first sentence of the introduction should be an unquestionable affirmation; therefore there is a need of complementing this first sentence with more references.

Reply: Necessary changes made in punctuations and reference added in first sentence.

  1. In papers standard writing using SI units, the unit is separated of the number by a space. This rule is only an exception for angles degrees and time nomenclature using prima.

Reply: Spacing provided

  1. The first time that cementoenamel junction appears, there is no reference to its abbreviation (CEJ). Then in line 103, the abbreviation was written without mention before the term. 

Reply: Abbreviations corrected

  1. Replace inside margin Exclusion criteria.

Reply: Replaced

  1. Line 151: Postoperative Instructions or Post‐operative instructions?

Reply: Changed to Post-Operative

  1. Integrate the sentence of lines 152/153 with the information of lines 154, 155 and 156?

Reply: Integrated

  1. Please try to standardize pre-operative or preoperative.

Reply: Standardized accordingly

  1. Shouldn't lines 184 to 186, be placed on 172 to 174?

Reply:  Statistical analysis was carried out following follow-up visits.

  1. I feel the need to been described how is evaluated the mean oral hygiene index‐simplified (OHI‐S) score. is this score in a range of 0 to 5, 0-10, 0-20 ? Which are the criteria of oral hygiene index‐simplified (OHI‐S) score?

Reply: Explanation provided

  1. I understand that discussion brings some topics that should be carried out in introduction, as example do a revision on similar membranes existing in market and why you choose Healiguide?

Reply: necessary changes made

  1. Also in conclusions, I consider it should point out parcial conclusions of the results, in order to offer to the readers a board perspective of the main conclusions of the work.

Reply: Necessary changes made

  1. I understood that alveolar bone level (ABL in mm) is measured on the radiographic images, but how is measured clinical attachment level (CAL in mm)? And probing pocket depth (PPD in mm). Please explain OHI-S, CAL and PPD, procedure. The only procedure I understood was ABL, which was correctly explained.

Reply: explanation provided

Reviewer 2:

several points are unclear in the manuscript:

  • Patient with pore oral hygiene was excluded. Did the authors set up any tracsable/defined measures to define/describe the exclusion?

Reply: Criteria provided

  • collagen is widely accepted as pro-healing matrix for a long time. Are the results presented in the manuscript surprising in that the RCM treatment group showed better outcomes?

Reply: Explanation provided

  • Patients showed a slight difference during the first 90 days and a significant difference after 180 days. How did the authors conclude that this difference was caused by RCM treatment? 

Reply: The difference seen on 180th day was present only in participants treated with RCM whereas the untreated group did not produce any significant change, So this change could be attributed to the RCM

Reviewer 3:

  1. The authors need to highlight better the novelty of this work compared to what has been found in literature. The use of collagen membranes is not a novel strategy.

Reply: Highlighted

  1. The authors should better characterize the collagen membranes used in this study (porosity, etc...)

Reply: Characteristics mentioned

Reviewer 2 Report

several points are unclear in the manuscript:

(1) Patient with pore oral hygiene was excluded. Did the authors set up any tracsable/defined measures to define/describe the exclusion?

(2) collagen is widely accepted as pro-healing matrix for a long time. Are the results presented in the manuscript surprising in that the RCM treatment group showed better outcomes?

(3) Patients showed a slight difference during the first 90 days and a significant difference after 180 days. How did the authors conclude that this difference was caused by RCM treatment? 

Author Response

Reviewer 1:

  1. In the introduction and discussion, there are several examples of ponctuation lack before references, and space between words and references is not well defined.  Also, the first sentence of the introduction should be an unquestionable affirmation; therefore there is a need of complementing this first sentence with more references.

Reply: Necessary changes made in punctuations and reference added in first sentence.

  1. In papers standard writing using SI units, the unit is separated of the number by a space. This rule is only an exception for angles degrees and time nomenclature using prima.

Reply: Spacing provided

  1. The first time that cementoenamel junction appears, there is no reference to its abbreviation (CEJ). Then in line 103, the abbreviation was written without mention before the term. 

Reply: Abbreviations corrected

  1. Replace inside margin Exclusion criteria.

Reply: Replaced

  1. Line 151: Postoperative Instructions or Post‐operative instructions?

Reply: Changed to Post-Operative

  1. Integrate the sentence of lines 152/153 with the information of lines 154, 155 and 156?

Reply: Integrated

  1. Please try to standardize pre-operative or preoperative.

Reply: Standardized accordingly

  1. Shouldn't lines 184 to 186, be placed on 172 to 174?

Reply:  Statistical analysis was carried out following follow-up visits.

  1. I feel the need to been described how is evaluated the mean oral hygiene index‐simplified (OHI‐S) score. is this score in a range of 0 to 5, 0-10, 0-20 ? Which are the criteria of oral hygiene index‐simplified (OHI‐S) score?

Reply: Explanation provided

  1. I understand that discussion brings some topics that should be carried out in introduction, as example do a revision on similar membranes existing in market and why you choose Healiguide?

Reply: necessary changes made

  1. Also in conclusions, I consider it should point out parcial conclusions of the results, in order to offer to the readers a board perspective of the main conclusions of the work.

Reply: Necessary changes made

  1. I understood that alveolar bone level (ABL in mm) is measured on the radiographic images, but how is measured clinical attachment level (CAL in mm)? And probing pocket depth (PPD in mm). Please explain OHI-S, CAL and PPD, procedure. The only procedure I understood was ABL, which was correctly explained.

Reply: explanation provided

Reviewer 2:

several points are unclear in the manuscript:

  • Patient with pore oral hygiene was excluded. Did the authors set up any tracsable/defined measures to define/describe the exclusion?

Reply: Criteria provided

  • collagen is widely accepted as pro-healing matrix for a long time. Are the results presented in the manuscript surprising in that the RCM treatment group showed better outcomes?

Reply: Explanation provided

  • Patients showed a slight difference during the first 90 days and a significant difference after 180 days. How did the authors conclude that this difference was caused by RCM treatment? 

Reply: The difference seen on 180th day was present only in participants treated with RCM whereas the untreated group did not produce any significant change, So this change could be attributed to the RCM

Reviewer 3:

  1. The authors need to highlight better the novelty of this work compared to what has been found in literature. The use of collagen membranes is not a novel strategy.

Reply: Highlighted

  1. The authors should better characterize the collagen membranes used in this study (porosity, etc...)

Reply: Characteristics mentioned

Editor: Please provide a simple summary according to the journal template (about
200 words)

Reply: As advised by the editor, simple summary is added

Reviewer 3 Report

The authors need to highlight better the novelty of this work compared to what has been found in literature. The use of collagen membranes is not a novel strategy.

The authors should better characterize the collagen membranes used in this study (porosity, etc...).

Author Response

(The authors gave the same response as above.)

Round 2

Reviewer 1 Report

The authors perform the corrections. Therefore, I accept the manuscript in the present form to be published.

Reviewer 2 Report

Maybe it will be more conclusive if more time points between 0-90, 90-180 can be added.

Reviewer 3 Report

The authors have addressed the comments made by the reviewer.